# Phonon Anharmonicity and Spin–Phonon Coupling in CrI_3_

**DOI:** 10.3390/ma16144909

**Published:** 2023-07-09

**Authors:** Luca Tomarchio, Lorenzo Mosesso, Salvatore Macis, Loi T. Nguyen, Antonio Grilli, Martina Romani, Mariangela Cestelli Guidi, Robert J. Cava, Stefano Lupi

**Affiliations:** 1Department of Physics, Sapienza University, Piazzale Aldo Moro 5, 00185 Rome, Italy; luca.tomarchio@uniroma1.it (L.T.);; 2INFN Section of Rome, P.Le Aldo Moro, 2, 00185 Rome, Italy; 3INFN—Laboratori Nazionali di Frascati, Via Enrico Fermi 54, 00044 Rome, Italy; 4Department of Chemistry, Princeton University, Princeton, NJ 08544, USA

**Keywords:** optical spectroscopy, van der Waals ferromagnet, spin–phonon coupling, phonon anharmonicity

## Abstract

We report on the far-infrared, temperature-dependent optical properties of a CrI${}_{3}$ transition metal halide single crystal, a van der Waals ferromagnet (FM) with a Curie temperature of 61 K. In addition to the expected phonon modes determined by the crystalline symmetry, the optical reflectance and transmittance spectra of CrI${}_{3}$ single crystals show many other excitations as a function of temperature as a consequence of the combination of a strong lattice anharmonicity and spin–phonon coupling. This complex vibrational spectrum highlights the presence of entangled interactions among the different degrees of freedom in CrI${}_{3}$.

## 1. Introduction

Three-dimensional van der Waals (vdW) layered crystals, like transition metal dichalcogenides [1,2] and halides [3], have attracted a great interest since they preserve a rich 2D physical phenomenology in combination with having applications in bulk devices [4,5,6]. Indeed, novel functional properties of these compounds can be found in their non-trivial electronic behaviors like excitonic interactions and dynamics [7] and spin/valley physics [8,9]. In particular, transition metal halides like CrI${}_{3}$ and CrCl${}_{3}$ have been found to host an intrinsic ferromagnetic order [10,11,12], which supports novel phases of matter, like the quantum Hall effect (QHA) [13,14], the spin liquid state [15,16], and the appearance of multiple magneto-optical effects [17,18,19]. Moreover, VdW magnets can be capitalized as interfacial layers, substrates, and tunnel barriers for spintronic applications and magnetic proximity effects [20,21].

Bulk CrI${}_{3}$ is a layered c-axis anisotropic ferromagnetic insulator with a Curie temperature T${}_{c}$ of 61 K. Below ∼220 K, a first-order structural phase transition transforms the unit cell from a monoclinic phase to rhombohedral layer stacking [22]. In each layer, the Cr atoms form a honeycomb structure surrounded by six iodine atoms in an octahedral coordination [23]. CrI${}_{3}$ has been theoretically predicted [24,25] to host strong interactions among phononic, electronic, and magnetic degrees of freedom, including a strong spin–orbit coupling (SOC) with the appearance of exotic quasi-particles like polarons [20,26,27,28], nonreciprocal magneto-electric effects [21,29,30], and topological spin waves [31]. A strong interplay between the lattice collective modes, the magnetic phase, and the electronic band structure emerges in CrI${}_{3}$ [32], as seen, for instance, in the significant hardening of the optical band gap below the Curie temperature [33].

Due to these different degree of freedom interactions, and the possibility to exploit them for nanoscale electro-optical, spintronic, and caloritronic applications, a study of the lattice anharmonicity and the corresponding energy relaxation is required. Indeed, the electron–phonon (e–ph) and phonon–phonon (ph–ph) interactions are key contributions to the electrical transport and the study of their correlations to magnetic and electronic degrees of freedom is absent in the experimental literature. The e–ph interaction is proportional to the number of phonons. The latter is determined by the balance between phonon generation and their thermalization characteristic times [34]. When the ph–ph interactions are not efficient enough to thermalize the phonons back to the equilibrium Bose–Einstein distribution, a non-thermal phonon population (hot phonons) can build up, eventually limiting the electronic transport properties of the system. This is especially true in 2D materials, where the reduced dimensionality decreases the density of decay channels [35].

This work addresses the infrared spectrum of a high-quality 300 μm thick CrI${}_{3}$ single crystal. In particular, we measure the whole phonon spectrum of CrI${}_{3}$ by crossing the structural and magnetic phase transitions down to the liquid helium temperature. In addition to the phonon modes predicted by the crystalline symmetry, we highlight the presence of many other vibrational modes associated with phonon overtones and combination bands and induced by anharmonicity. In particular, we report on the appearance of an anomalous linewidth behavior for an $A_{1u}$ vibration near 135 cm${}^{-1}$, suggesting a coupling with a background of spin excitations at lower energies. We also investigate the temperature dependence of a strong $E_{u}$ optical phonon near 230 cm${}^{-1}$ by studying its contribution to optical reflectance. Finally, we analyze the frequency softening and linewidth broadening of all vibrational modes versus (vs.) temperature, associating them with the presence of three- and four-phonon anharmonic processes.

## 2. Materials and Methods

CrI${}_{3}$ single crystals were synthesized by a chemical vapor deposition technique. A 1 g mixture of the stoichiometric ratio of Cr metal and I${}_{2}$ pieces (Alfa Aesar, 99.99%) was packed in an evacuated quartz glass tube (22 cm long and 16 mm wide) and heated in a three-zone furnace, set at zone temperatures 650, 550, and 600 °C, for one week. Finally, the “charge” was placed in the 650 °C zone. Many CrI${}_{3}$ crystals were grown in the 550 °C zone as millimeter-sized grey plates with variable thicknesses and comparable physical properties. The crystals were stable in the air for a few hours. A single flake 300 μm thick was used for the entire experiment. It was sealed in a vacuum environment. No degradation was observed during the optical measurements.

Optical measurements were performed through a Bruker (Billerica, MA 01821, USA) Vertex 70v Infrared interferometer, coupled with different detectors and beamsplitters covering the spectral region from terahertz (THz, 20 cm${}^{-1}$) to near-infrared (NIR, 15,000 cm${}^{-1}$). A liquid-He-cooled bolometer was used for measurements from 20 cm${}^{-1}$ up to 600 cm${}^{-1}$, while a pyroelectric detector working at room T was used for the higher frequencies. Optical measurements were taken at various temperatures through a He-cooled JANIS (ST-100-FTIR) cryostat from Lake Shore Cryotronics (Westerville, OH 43082, USA).

### Models for Anharmonic Phonon Scattering

The linewidth and frequency variations of the vibrational modes with temperature were examined through the theory of phonon–phonon interactions and thermal expansion of the lattice [36,37]. As a first step, the phonon frequencies and linewidths were extracted through a fit of the frequency-dependent dielectric function using a Lorentz model:(1)$$\epsilon \left(\omega \right)=\epsilon_{\infty }+\sum_{j}\frac{\nu_{pj}^{2}}{\left(\nu_{0j}^{2}-\nu^{2}\right)-i\gamma_{j}\nu }$$
where $\nu_{0j}$ is the central frequency of the *j*-th phonon, $\gamma_{j}$ is the linewidth, and $\nu_{pj}$ is the corresponding amplitude.

The frequency shift Δν of phonons at constant pressure arises from a pure-volume contribution ${\left(\Delta \nu \right)}_{latt}$ resulting from thermal expansion, plus some pure-temperature contributions ${\left(\Delta \nu \right)}_{anh}$ given by phonon–phonon scattering anharmonicities [37,38,39].
(2)$$\Delta \nu ={\left(\Delta \nu \right)}_{latt}+{\left(\Delta \nu \right)}_{anh}$$

The cubic and quartic anharmonic terms can be described including contributions from three- and four-phonon anharmonic processes. Equation (Equation 2) can be written as [38,39]
(3)$$\Delta \nu =\nu_{0}\left(e^{-\gamma \beta T}-1\right)+A\left[1+\frac{2}{e^{h\nu_{0}/2k_{B}T}-1}\right]+B\left[1+\frac{3}{e^{h\nu_{0}/3k_{B}T}-1}+\frac{3}{{\left(e^{h\nu_{0}/3k_{B}T}-1\right)}^{2}}\right]$$
where $\nu_{0}$ is the zero temperature frequency, approximated by the 5 K experimental value in the fitting process. The coefficient γβ is given by the product of the Grüneisen parameter γ and the volume thermal expansion coefficient β, while *A* and *B* are coefficients that weight the three- and four-phonon anharmonic processes, respectively.

A similar result can be written for the linewidth behavior of the vibrational modes. Since the ph–ph scattering processes are sensible to the phonon population, this induces a dependence of the linewidth of the single modes on temperature. The resulting relation follows the same analytical form of Equation (Equation 3), but with the benefit of having a null dependence on the quasi-harmonic term induced by lattice dilation
(4)$$\Gamma \left(T\right)=\Gamma_{0}+C\left[1+\frac{2}{e^{h\nu_{0}/2k_{B}T}-1}\right]+D\left[1+\frac{3}{e^{h\nu_{0}/3k_{B}T}-1}+\frac{3}{{\left(e^{h\nu_{0}/3k_{B}T}-1\right)}^{2}}\right]$$
where $\Gamma_{0}$ is the linewidth at T=0 K and the coefficients *C* and *D* are constants characterizing the contribution of the three- and four-phonon processes to the linewidth, respectively.

## 3. Results

The crystal structure of CrI${}_{3}$ is shown in the inset of Figure 1b (see Methods for the growing process description). The chromium (Cr) and iodine (I) atoms form honeycomb ordered layers. The arrows in the figure indicate the crystal axes. The bulk crystal structure of CrI${}_{3}$ at room temperature is described by a monoclinic (space group C2/m) unit cell. Below the structural phase transition around ∼220 K, this changes to a rhombohedral symmetry (space group R$\bar{3}$) [22].

In Figure 1a, we report the transmittance T(ν) and the reflectance R(ν) vs. frequency (ν) at 300 K of a high-quality 300μm thick CrI${}_{3}$ single crystal. Figure 1b,c shows T(ν) and R(ν) (the latter is limited to the strong phonon mode) at various temperatures. While R(ν) is dominated by a strong phonon absorption around 230 cm${}^{-1}$, T(ν) shows many minima below and above the main phonon mode at 230 cm${}^{-1}$. Finally, in Figure 1d, we show the absorption coefficient as extracted through a Kramers–Kronig consistent fitting process of R and T [40], using a combination of Lorentzian functions to reproduce the complex dielectric function expressed by Equation (Equation 1).

A plethora of low energy absorption peaks can be observed in Figure 1d, which can be associated with both single- and multi-phonon excitations. Vibrational excitations up to the strong absorption at about 230 cm${}^{-1}$ ($E_{u}$ symmetry, mainly due to Cr vibrations) are predicted by the $D_{3d}$ point group symmetry [24,41], describing the single CrI${}_{3}$ monolayer. Here, the phonon spectrum allows five IR active transitions, namely three $E_{u}$ modes and two $A_{2u}$ modes, three inactive modes (one $A_{1u}$ and two $A_{2g}$), and six Raman-active modes (two $A_{1g}$ and four $E_{g}$). This 2D description fits the bulk response well due to the van der Waals nature of the crystal and the in-plane polarization of the incident radiation. At higher frequencies, a series of sharp peaks can be seen in Figure 1c, with a strong spectral weight from 300 to 360 cm${}^{-1}$. These higher frequency excitations are not predicted by the $D_{3d}$ point group symmetry or by the ab initio results for CrI${}_{3}$ [24,42] and are associated instead with a combination of Raman and IR fundamental modes mainly as a consequence of strong anharmonic effects. In the following section, we will discuss the strong T dependence of the IR absorption coefficient of CrI${}_{3}$ in terms of phonon–phonon (ph–ph) and phonon–spin (ph–spin) interactions.

## 4. Discussion

The transparency increase in the phonon region with decreasing temperature, as seen in Figure 1c, suggests anharmonicity plays an important role in the vibrational spectrum of CrI${}_{3}$ [43,44,45]. To analyze these anharmonic effects, we investigate as a function of temperature the central frequency shift and linewidth variation of the main phonon peaks dominating the absorption spectrum (as observed in Figure 1). These peaks are indexed in terms of their central frequency (at 300 K) and symmetry [33]: P1 (82 cm${}^{-1}$, $E_{u}$), P2 (113 cm${}^{-1}$, $E_{u}$), P3 (133 cm${}^{-1}$, $A_{2u}$), P4 (276 cm${}^{-1}$, $E_{u}+E_{g}$), P5 (291 cm${}^{-1}$, $E_{g}+A_{2u}$), P6 (310 cm${}^{-1}$, $E_{u}+A_{1g}$), P7 (326 cm${}^{-1}$, $E_{u}+E_{g}$), P8 (337 cm${}^{-1}$, $E_{u}+E_{g}$), P9 (347 cm${}^{-1}$, $E_{u}+E_{g}$), Q1 (224.5 cm${}^{-1}$, $E_{u}$), Q2 (231.5 cm${}^{-1}$, $E_{u}$). We show the phonon frequency and linewidth temperature dependence as obtained by the fitting procedure (see above) in Figure 2 and Figure 3 (notice that the color code in Figure 2 and Figure 3 is the same in Figure 1). Figure 2a shows the contribution of the three lowest-frequency phonons to the absorption coefficient. Their central frequency softens with temperature as shown in Figure 2b. These values have been subtracted by their corresponding quantities at 5 K and each curve is shifted by a vertical offset for the sake of clarity. The jump in the central frequency of peak P1 (blue) and P3 (green) between 250 K and 200 K can be associated with the structural first-order transition of CrI${}_{3}$, which converts the crystal structure from monoclinic to rhombohedral, affecting both the in-plane and out-of-plane atomic distances [22]. The peak P2 instead is barely affected by the transition.

When looking at the temperature dependence of P1, P2, and P3 linewidths in Figure 2c, it is possible to highlight three different behaviors. The P1 linewidth (blue line) is practically constant with temperature, indicating a weak ph–ph interaction, as expected for the lowest energy modes, due to the reduced density of available decay channels. P2 (red line), instead, shows a minor increase in the linewidth with temperature. This is a fingerprint of weak anharmonicity due to ph–ph interactions since multiple phonon scattering increases at higher temperatures due to the growing phonon population. P3 (green line) follows instead an opposite behavior with respect to P2: its linewidth decreases with temperature. In particular, this peak practically disappears from the absorption spectrum below the Curie temperature and its fitting parameters cannot be extracted below 50 K. The strong intensity reduction of this phonon mode has been theoretically predicted due to strong spin–phonon coupling in the magnetic phase [32].

Above the magnetic transition, the P3 linewidth decreases at increasing temperatures. This effect cannot be explained in terms of a conventional ph–ph scattering process. The linewidth temperature dependence suggests instead the presence of a further magnetic scattering channel surviving above the Curie temperature, as found in the iso-structural α-RuCl${}_{3}$ material. Here, a broad magnetic continuum survives up to 100 K [45,46], well above the magnetic ordering temperature of 14 K.

Near 230 cm${}^{-1}$, two vibrational modes (Q1 and Q2) can be observed in the reflectivity spectrum at all temperatures (see Figure 1c). This double structure is reflected in the absorption coefficient (see Figure 2d), being characterized by a main peak around 225 cm${}^{-1}$ and a broad shoulder at about 235 cm${}^{-1}$. From symmetry arguments, only an $E_{u}$ mode is expected by the $D_{3d}$ point group. Therefore, this doublet could be related to the splitting of the $E_{u}$ double degenerate vibration due to local crystal distortions [47]. DFT+SOC calculations [24] also suggest a degeneracy splitting. However, the expected frequency separation is predicted to be smaller than 1 cm${}^{-1}$ both in the ferromagnetic and anti-ferromagnetic ground state. This is in contrast to the frequency separation (nearly 7 cm${}^{-1}$) observed experimentally, and nearly constant with temperature. Therefore, such a larger frequency separation would suggest a strong SOC in CrI${}_{3}$, in agreement with ref. [48]. A measurement of the monolayer CrI${}_{3}$ infrared activity would shine further light on the origin of this splitting.

The frequency shift and linewidth behavior vs. T of the Q1 and Q2 modes are shown in Figure 2e,f, respectively. The linewidths for both modes follow a decreasing behavior for decreasing temperatures down to nearly 100 K. Below this temperature, the linewidth T dependence is inverted. A similar trend has been observed for the reststrahlen band in α-RuCl${}_{3}$ [45], and it has been linked to its stacking sequence. In particular, the stacking order and stacking sequence determine the overall reflectance in α-RuCl${}_{3}$, actually boosting its value by a factor of two across the structural transition. However, this is not the case for CrI${}_{3}$, where the reflectivity enhancement across the structural transition is merely 10%, suggesting a different origin for the linewidth variation with temperature of Q1 and Q2 modes. The increasing behavior of the Q1 and Q2 linewidth below 100 K could be explained instead (as for the P3 mode) as the appearance of a new scattering channel related to the magnetic order.

### Phonon Anharmonicity

The blue shift of Q1 and Q2 modes with temperature (see Figure 2e) can be reproduced through the thermal expansion of the lattice and the theory of phonon–phonon interactions, as described by Equation (Equation 3) (see Methods). This equation contains three temperature-dependent frequency shift terms: a first quasi-harmonic term depending on the volume thermal expansion coefficient β and the Grüneisen constant γ, and the three- and four-phonon anharmonic processes, respectively, weighted by two free coefficients *A* and *B*.

From the knowledge of the thermal expansion coefficient and the Grüneisen constants for the CrI${}_{3}$ lattice vibrations [32], the quasi-harmonic (first) term in Equation (Equation 3) has a negligible effect on the blue shift of Q1 and Q2. Therefore, the observed blue shift is practically determined by the four- and four-phonon scattering anharmonic terms. Equation (Equation 3)’s coefficients, as obtained by the Q1 and Q2 data fitting, are shown in Table 1 and in Figure 2e as colored dashed lines. The four-phonon scattering processes contribute strongly to both Q1 and Q2, while the three-phonon scattering processes were found to be non-negligible only for Q2.

Similarly to the Q1 and Q2 modes, blue shifts with decreasing temperature can be observed also for the higher energy modes seen in Figure 3a,d. In Table 1, we report the coefficients A and B as obtained from the best fit of their frequency shifts (dashed lines in Figure 3b,e). These results highlight a dominant contribution given by the three-phonon scattering processes, as expected for high-frequency modes due to the high phonon DOS at lower energies participating in the decay mechanism [24]. An analysis of the P4, P5, and P6 mode linewidths reveals instead no relevant modifications across the temperature range (see Figure 3c), which can be explained by the small linewidth of the parent Q1 and Q2 modes with respect to the P4, P5, and P6 ones (these latter vibrations are described by a linear superposition of Raman and IR active modes). A similar fit procedure of Equation (Equation 3) can instead be applied for the P7, P8, and P9 mode linewidths, as shown in Figure 3f, using a similar temperature-dependent model (see Equation (Equation 4) in Methods).

## 5. Conclusions

We reported on the far-infrared optical response of a CrI${}_{3}$ van der Waals ferromagnet single crystal at various temperatures, investigating its complex phononic absorption spectrum and highlighting its temperature dependence in terms of strong lattice anharmonicities. Our data reveal significant mode softening and linewidth changes with temperature, described in terms of multi-phonon and spin–phonon scattering processes. We highlight the anomalous behavior of the vibrational modes at 133 cm${}^{-1}$ ($A_{2u}$) and 230 cm${}^{-1}$ ($E_{u}$), suggesting a spin–phonon coupling surviving at temperatures above the magnetic transition. These results mark CrI${}_{3}$ as a material hosting strong phonon interactions, suggesting a complex interplay between the lattice, magnetic, and electronic degrees of freedom that should be studied as a function of time in future pump–probe measurements. Moreover, given its variable properties as a function of thickness, this same investigation should be extended to few-layered CrI${}_{3}$ (and other VdW systems) in future works.

## Figures and Tables

**Figure 1 materials-16-04909-f001:**
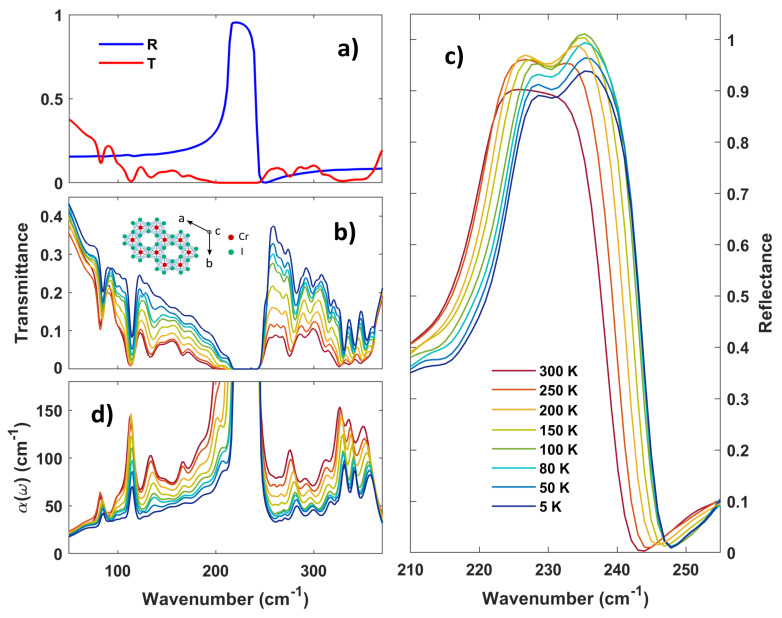
**Far infrared optical properties of a CrI${}_{3}$ single crystal.** (**a**) Far infrared reflectance (R) and transmittance (T) at 300 K for a CrI${}_{3}$ single crystal with a thickness of 300 μm. (**b**) Optical transmittance at different temperatures for a 300 μm thick CrI${}_{3}$ single crystal. Inset: top view of the crystal structure of CrI${}_{3}$. The Cr and I atoms form honeycomb ordered layers. Arrows indicate the crystal axes. (**c**) Optical reflectance of a 300 μm thick CrI${}_{3}$ at different temperatures. R(ν) is dominated by a strong phonon mode near 230 cm${}^{-1}$. (**d**) Absorption coefficient at various temperatures as extracted from the transmittance measurements fitting process. An increased transparency is observed at decreasing temperatures.

**Figure 2 materials-16-04909-f002:**
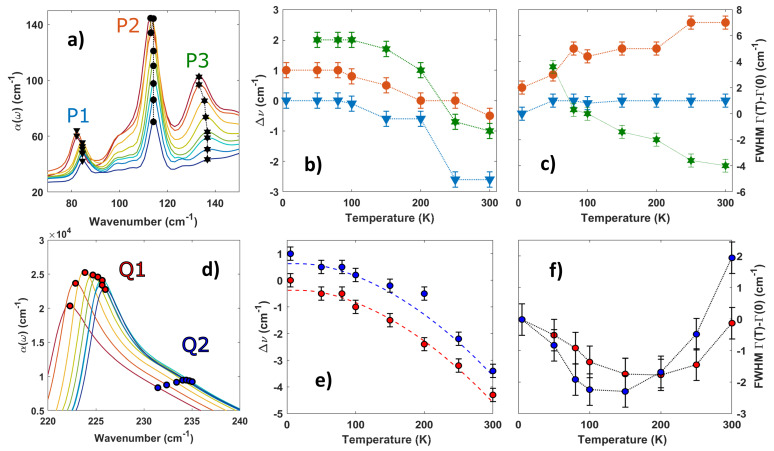
**Frequency shift and linewidth variation of the CrI${}_{3}$ vibrational modes with temperature.** (**a**) Depiction of three low-frequency modes in the IR absorption spectrum. The black points highlight the temperature evolution. (**b**,**c**) Central frequency and linewidth evolution of the P1, P2, and P3 phonon modes. Dotted lines are a guide for the eye. The values have been plotted as a difference with respect to the 5 K (50 K for P3) value and shifted by an offset for clarity. Error bars are extracted from the best fitting process. (**d**) Absorption coefficient highlighting the Q1 and Q2 phonon modes at varying temperatures. (**e**,**f**) Central frequency and linewidth evolution of the Q1 and Q2 phonon modes. The values for the frequency shift have been plotted as the difference with respect to the 5 K value and shifted by an offset for clarity. Discontinuous lines in (**e**) highlight the best fitting procedure in accordance to Equation (Equation 3).

**Figure 3 materials-16-04909-f003:**
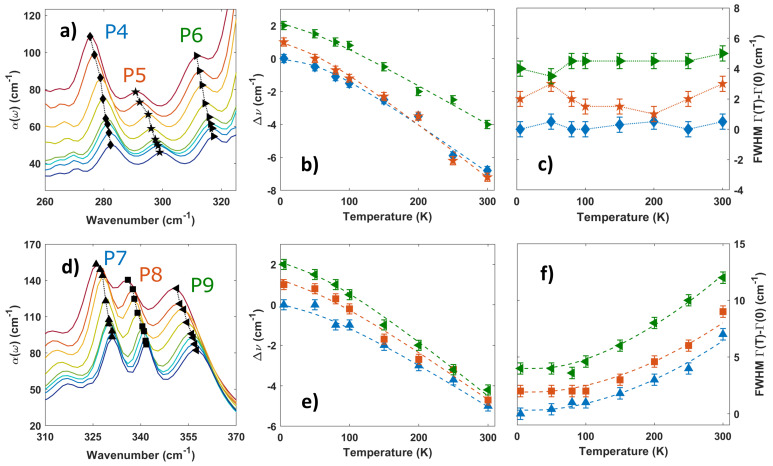
**Frequency shift and linewidth variation of the CrI${}_{3}$ vibrational modes with temperature.** (**a**) Depiction of three frequency modes in the THz absorption spectrum: P4, P5, and P6. (**b**,**c**) Frequency and linewidth evolution with temperature (with respect to 5 K) of the vibrational modes P4, P5, and P6. The values are plotted as the difference with respect to the 5 K value and shifted by an offset for clarity. The frequency shifts were fitted (discontinuous lines) by a model for the anharmonic three- and four-phonon scattering processes (Equation (Equation 3)). Error bars are extracted from the best fitting process of the Lorentzian function. The linewidth modulations are approximately constant across the temperature range. (**d**) Depiction of the three highest frequency modes in the FIR absorption spectrum: P7, P8, and P9. (**e**,**f**) Frequency and linewidth variation with temperature for P7, P8, and P9 phonon modes. The values are plotted as the difference with respect to the 5 K value and shifted by an offset for clarity. Discontinuous lines highlight the best fitting procedure in accordance with Equations (Equation 3) and (Equation 4).

**Table 1 materials-16-04909-t001:** Fitting parameters A and B appearing in Equation (Equation 3) for the vibrational modes reported in Figure 2 and Figure 3. The reported δA and δB uncertainties are extracted by the best fitting process using a Least Absolute Residuals (LAR) algorithm.

	P4	P5	P6	P7	P8	P9	Q1	Q2
*A* (cm${}^{-1}$)	−2.7	−3.6	−3	−1.8	−2.2	−3.7	0	−0.20
*B* ($\times 10^{-2}$ cm${}^{-1}$)	−4.5	−1.3	−1	−18	−20	−5	−12	−18
δA (cm${}^{-1}$)	0.5	0.7	0.6	0.4	0.4	0.7	0	0.04
δB ($\times 10^{-2}$ cm${}^{-1}$)	0.9	0.3	0.2	4	4	−1	−3	4

## Data Availability

Derived data supporting the findings of this study are available from the corresponding author upon request.

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
