# Peer review of "Phonon Anharmonicity and Spin–Phonon Coupling in CrI3"

_materials, 2023, doi:10.3390/ma16144909_

Round 1

Reviewer 1 Report

The manuscript entitled “Materials-2409613” dealing with photons has been reviewed. The paper has been nicely written but needs some improvement. Please follow my comments.

1.     How authors counted the wavenumber in Fig 1? What was the devise for that?

2.     Authors are encouraged to state what industry benefits from this work.

3.     What is the main issue that will be solved by this investigation? Please clarify it in the text.

4.     Please add a brief statement on your methodology in the abstract.

5.     What is the future direction of this work?

6.     Please proofread the paper.

7.     Photons are the driving factor for lasers. Laser has many advantages over the conventional manufacturing method which can be highlighted in your paper. Please read the following manuscript and add it to the literature to show engineering application of photons. “Laser subtractive and laser powder bed fusion of metals: review of process and production features”

English needs some improvements. 

Author Response

We thank Reviewer 1 for the comments. However, our manuscript does not deal with photons but with phonons. In our paper, light, and thus photons, are only used as a probe for investigating the physical properties of CrI3.

We answer the questions raised by Reviewer 1 here below:

  1. To measure the transmittance and reflectance spectrum we used a Michelson interferometer as described in the methods section. Wavenumbers are extracted through a fast Fourier transform of the interferometer signal. This is done automatically by the instrument.
  2. We introduce the application of VdW magnets in the introduction section. They can be capitalized as substrates, interfacial layers, and tunnel barriers for magnetic proximity effects and spintronic applications. The industries that benefit from the study of these materials are the electronic and magnetic industries.
  3. There is no specific issue to solve in this paper. Our work reports on the observation of spin-phonon and phonon-phonon anharmonicities, whose study is a key ingredient in understanding the internal correlations of quantum materials like VdW magnets. Moreover, as we write in the introduction section, understanding phonon-phonon interaction provides information about hot phonons, which limit the electronic transport properties of 2D systems.
  4. We add a sentence in the abstract highlighting the study of the optical reflectance and transmittance spectrum as a function of temperature.
  5. This work will be of support for future studies of electron and thermal transport in CrI3 as a function of time, like optical pump-THz probe measurements. We add this point in the conclusion.
  6. Done
  7. The paper does not deal with photons, but phonons, as pointed out earlier. The sentence “Laser subtractive and laser powder bed fusion of metals: review of process and production feature” suggested by Reviewer 1 unfortunately has no correlation with the topics discussed in our work.

Reviewer 2 Report

The article presents an intriguing experimental analysis of anharmonism processes in CrI3. Nonetheless, some points should be described and discussed in more detail. In the case of this article, the reviewer must admit that they could not determine whether the interpretation of the measurements was accurate or inaccurate, and thus asks the authors to analyze this aspect once more within their team.

More specific comments are as follows:

1) The study is centered on the optical properties of CrI3, a single metal halide. It would have been beneficial to compare this with other similar materials to provide context to the reader and to understand the general processes of anharmonicity. Are similar effects and phase transitions observed in other halides?

2) You mentioned the phonon modes predicted by the crystalline symmetry. It would be helpful to provide this group theory analysis in a more explicit form in the paper or supplementary material, such as a table.

3) The authors mentioned that the CrI3 single crystals were synthesized via a chemical vapor transport technique (Cite: “Many CrI3 crystals were formed in the 550 C zone”). However, there's a lack of discussion regarding the scalability and reproducibility of this method. Can it produce consistent results when scaled up, and is it cost-effective? Are the samples identical? Is it controlled? The authors state that the crystals are stable in air for a few hours. Did you check sample degradation and its impact on the measurements?

4) While the authors mention potential applications in nanoscale electro-optical, spintronic, and caloritronic devices, it would be helpful if they discussed more about how these potential applications could be realized. This includes potential device designs, integration methods with existing technologies, or any foreseeable challenges.

5) The authors base their analysis of anharmonic effects on temperature variations. Could other factors also influence anharmonic effects?

6) The authors do not discuss potential limitations of their study.

7) There's no mention of any statistical analysis to validate the results or the uncertainty associated with the measurements. The authors mention a "20% relative uncertainty" from the fitting process but there is no detailed explanation of how this figure is derived. This could probably be added in the supplementary material.

7) As the authors write, "We highlight an anomalous behavior of the vibrational modes at 133 cm−1 (A2u) and 230 cm−1 (Eu), suggesting a spin-phonon coupling surviving for temperatures above the magnetic transition". Is this the only possible explanation for the observed effects? Having read the article, I could not come to a conclusion about the accuracy and correctness of this statement.

English if fine for me

Author Response

We thank Reviewer 2 for her/his positive comments about our manuscript. In the following we reply to all issue her/his raised.

1) The study is centered on the optical properties of CrI3, a single metal halide. It would have been beneficial to compare this with other similar materials to provide context to the reader and to understand the general processes of anharmonicity. Are similar effects and phase transitions observed in other halides?

Reply 1

CrI3 discussed in our work is part of a broader family of halides and 2D VdW magnets that possess similar properties. For instance, a ferromagnet ordering was found also in VI3 and CrCl3, as we report in the introduction section. However, the study of phonon anharmonicities and their correlations to magnetic and electronic degrees of freedom is absent in the experimental literature. Our manuscript fill this gap. We also compare our results to theoretical phononic calculations suggesting similar properties among different families and finding definitive matching features.

2) You mentioned the phonon modes predicted by the crystalline symmetry. It would be helpful to provide this group theory analysis in a more explicit form in the paper or supplementary material, such as a table.

Reply 2

In order to follow the Reviewer suggestion, we introduce in the main text a sentence listing all the phonon modes predicted by the group theory analysis.

3) The authors mentioned that the CrI3 single crystals were synthesized via a chemical vapor transport technique (Cite: “Many CrI3 crystals were formed in the 550 C zone”). However, there's a lack of discussion regarding the scalability and reproducibility of this method. Can it produce consistent results when scaled up, and is it cost-effective? Are the samples identical? Is it controlled? The authors state that the crystals are stable in air for a few hours. Did you check sample degradation and its impact on the measurements?

Reply 3

Samples come from the same batch, present the same physical and chemical properties. They vary in size and thickness. A single flake 300 um thick has been used for the entire experiment. It was sealed in a vacuum environment. No degradation has been observed during the optical measurements. Regarding the scalability of the growing method and its cost-effective, one can refer to Ref. 22 related to the growing process.

4) While the authors mention potential applications in nanoscale electro-optical, spintronic, and caloritronic devices, it would be helpful if they discussed more about how these potential applications could be realized. This includes potential device designs, integration methods with existing technologies, or any foreseeable challenges.

Reply 4

Our work deals with the study of anharmonic phonon interaction and spin-phonon interaction in CrI3. In particular, we suggest the importance of studying the CrI3 phonon population which generate strong constraints to electronic transport. The applications to spintronic, electro-optic devices, etc., are suggested by more general studies on these systems. See Ref. 4-6.

5) The authors base their analysis of anharmonic effects on temperature variations. Could other factors also influence anharmonic effects?

Reply 5

We thank Reviewer 2 for this comment. “Phonon anharmonicities” describe the interaction between multiple phonons. The variation in temperature is the most direct way to study this effect since the phonon population, and thus the degree of interaction, changes as a function of temperature. This is shown by Eq. 2 main text. Another possibility to modify the phonon interaction consists in applying a strong external pressure in the GPa range.  

6) The authors do not discuss potential limitations of their study.

Reply 6

VdW magnets like CrI3 have been shown to have variable properties as a function of thickness. Here, we are dealing with a 300 um single crystal, showing bulk properties. The extension of our investigation to few layers CrI3 (and other VdW systems) will concern a forthcoming paper.

7) There's no mention of any statistical analysis to validate the results or the uncertainty associated with the measurements. The authors mention a "20% relative uncertainty" from the fitting process but there is no detailed explanation of how this figure is derived. This could probably be added in the supplementary material.

Reply 7

The error bars extracted from the best-fitting process are highlighted in Table 1. The fitting is performed on MATLAB using a Least absolute residuals (LAR) method. The LAR method finds a curve that minimizes the absolute difference of the residuals. The uncertainty over the extracted coefficients is a direct result of the weights over the experimental data, which are shown in Fig. 2 and 3. We modify the text in Table 1 to address this concern.

8) As the authors write, "We highlight an anomalous behavior of the vibrational modes at 133 cm−1 (A2u) and 230 cm−1 (Eu), suggesting a spin-phonon coupling surviving for temperatures above the magnetic transition". Is this the only possible explanation for the observed effects? Having read the article, I could not come to a conclusion about the accuracy and correctness of this statement.

Reply 8

Our explanation is based on Ref. 32. Other possibilities based on phonon-phonon or e-ph interactions, can be discarded as they cannot take into account the thermal properties of the two modes.

Reviewer 3 Report

In this manuscript, the authors reported on the far-infrared optical response of a CrI3 van der Waals ferromagnet single crystal at various temperatures, investigating its complex phononic absorption spectrum and highlighting its temperature dependence in terms of strong lattice anharmonicities. The authors have done a good job in explaining in detail the results by providing sufficient data to support their claims. The data presented in this paper is highly credible. However, this manuscript has some minor mistakes as follows.

1)     Line 102, 108, 113 in Page 4, “Fig 1c” should be “Fig 1d”.

2)     Figure 2e and 2f, why there are no error bars?

Author Response

In this manuscript, the authors reported on the far-infrared optical response of a CrI3 van der Waals ferromagnet single crystal at various temperatures, investigating its complex phononic absorption spectrum and highlighting its temperature dependence in terms of strong lattice anharmonicities. The authors have done a good job in explaining in detail the results by providing sufficient data to support their claims. The data presented in this paper is highly credible. However, this manuscript has some minor mistakes as follows.

1)     Line 102, 108, 113 in Page 4, “Fig 1c” should be “Fig 1d”.

Reply 1

We thank Reviewer 3, we have fixed the typos.

2)     Figure 2e and 2f, why there are no error bars?

Reply 2

We have introduced the error bars also in Fig. 2e and 2f as extracted from the Lorentz fit.

Reviewer 4 Report

1. Author has mentioned vs. temperature in line no. 49. Author should be use appropriate symbol with abbreviation at first and after that use frequently.

2. Rephrase the line no. 50-54 in introduction section.

3. Author should be mention the mass of I2 pieces in line no. 57.

4. Introduce equation 4  at first and then use equation 2 in line no. 79.

5. What is the reason for showing the more deflection of P3 in Fig. 2 b at different temperature.

6. Provide the error bar value in tabulate form.

7. Author should be use reference no. 48 in manuscript.

The quality of English language of this manuscript having high level which is appropriate for this journal.

Author Response

1) Author has mentioned vs. temperature in line no. 49. Author should be use appropriate symbol with abbreviation at first and after that use frequently.

Reply 1

We thank Reviewer 3, we added versus (vs.) in the main text.

2) Rephrase the line no. 50-54 in introduction section.

Reply 2

We have changed the order of the results presentation in the main text.

3) Author should be mention the mass of I2 pieces in line no. 57.

Reply 3

The molar ratio between the chromium powder and iodine beads (99.999%) is 1:3

4) Introduce equation 4  at first and then use equation 2 in line no. 79.

Reply 4

We have shifted the equations order as suggested.

5) What is the reason for showing the more deflection of P3 in Fig. 2 b at different temperature.

Reply 5

The reason is given by the disappearance of the peak at 5 K. The fitting process is not able to give a reliable estimate of the phonon properties given the strong broadening. We add a sentence in Fig. 2 caption.

6) Provide the error bar value in tabulate form.

Reply 6

We have added two rows in Table 1 addressing the parameters uncertainty.

7) Author should be use reference no. 48 in manuscript.

Reply 7

We fixed this issue.

Round 2

Reviewer 1 Report

The paper is ready to publish.

Author Response

We thank reviewer 1 to agree on publication.

Reviewer 2 Report

Thank you for the significant improvement to the manuscript. I believe that a some parts of the response to the reviewer should be directly incorporated into the text.

Initial discussion 1. The study is centered on the optical properties of CrI3, a single metal halide. It would have been beneficial to compare this with other similar materials to provide context to the reader and to understand the general processes of anharmonicity. Are similar effects and phase transitions observed in other halides?

Reply 1: CrI3 discussed in our work is part of a broader family of halides and 2D VdW magnets that possess similar properties. For instance, ferromagnetic ordering was also found in VI3 and CrCl3, as we report in the introduction section. However, the study of phonon anharmonicities and their correlations to magnetic and electronic degrees of freedom is absent in the experimental literature. Our manuscript fills this gap. We also compare our results to theoretical phononic calculations suggesting similar properties among different families and finding definitive matching features.

----------------------------------------

I suggest that the response to the reviewer should be incorporated into the introduction for better understanding of the work's motivation and what it contributes (the study of phonon anharmonicities and their correlations to magnetic and electronic degrees of freedom is absent in the experimental literature).

----------------------------------------

Initial discussion  3. The authors mentioned that the CrI3 single crystals were synthesized via a chemical vapor transport technique (Cite: “Many CrI3 crystals were formed in the 550 C zone”). However, there's a lack of discussion regarding the scalability and reproducibility of this method. Can it produce consistent results when scaled up, and is it cost-effective? Are the samples identical? Is it controlled? The authors state that the crystals are stable in air for a few hours. Did you check sample degradation and its impact on the measurements?

Reply 3: The samples come from the same batch and present the same physical and chemical properties. They vary in size and thickness. A single flake 300 μm thick has been used for the entire experiment. It was sealed in a vacuum environment. No degradation has been observed during the optical measurements. Regarding the scalability of the growing method and its cost-effectiveness, one can refer to Ref. 22 related to the growing process.

----------------------------------------

I suggest that the synthesis should be described in more detail.

----------------------------------------

Initial discussion  6. The authors do not discuss potential limitations of their study.

Reply 6: VdW magnets like CrI3 have been shown to have variable properties as a function of thickness. Here, we are dealing with a 300 μm single crystal, showing bulk properties. The extension of our investigation to few-layered CrI3 (and other VdW systems) will be addressed in a forthcoming paper.

----------------------------------------

I suggest that the limitations should be explicitly described in the conclusions.

After this modification and additional control paper can be published. 

Author Response

We thank reviewer 2 for highlighting this missing information in the main text.

Round 2: answer 1

I suggest that the response to the reviewer should be incorporated into the introduction for better understanding of the work's motivation and what it contributes (the study of phonon anharmonicities and their correlations to magnetic and electronic degrees of freedom is absent in the experimental literature).

Round 2: Reply 1

We have introduced the sentence in the text as suggested by reviewer 2 (blue text).

Round 2: Answer 3

I suggest that the synthesis should be described in more detail.

Round 2: Reply 3

We have improved the synthesis section (blue text) as suggested by reviewer 2 by introducing the samples properties and the comments in reply 3 (round 1).

Round 2: Answer 6)

I suggest that the limitations should be explicitly described in the conclusions.

Round 2: Reply 6

We add a sentence in the conclusions as suggested by reviewer 2 (blue text).